# Ethnoveterinary Potential of *Acacia* (*Vachellia* and *Senegalia*) Species for Managing Livestock Health in Africa: From Traditional Uses to Therapeutic Applications

**DOI:** 10.3390/plants14193107

**Published:** 2025-10-09

**Authors:** Nokwethemba N. P. Msimango, Adeyemi O. Aremu, Stephen O. Amoo, Nqobile A. Masondo

**Affiliations:** 1Agricultural Research Council-Vegetable, Industrial and Medicinal Plants, KwaMhlanga/Moloto Road (R573), Roodeplaat, Private Bag X293, Pretoria 0001, South Africa; msimangon@arc.agric.za (N.N.P.M.); amoos@arc.agric.za (S.O.A.); 2Indigenous Knowledge Systems Centre, Faculty of Natural and Agricultural Sciences, North-West University, Private Bag X2046, Mmabatho 2745, South Africa; oladapo.aremu@nwu.ac.za; 3School of Life Sciences, College of Agriculture, Engineering and Science, University of KwaZulu-Natal, Private Bag X54001, Durban 4000, South Africa; 4Unit for Environmental Sciences and Management, Faculty of Natural and Agricultural Sciences, North-West University, Potchefstroom 2531, South Africa

**Keywords:** antibacterial, anti-inflammatory, antioxidant, biodiversity, fabaceae, phytochemicals

## Abstract

In Africa, the folkloric practices involving plant-based remedies play a crucial role in livestock farming, often attributed to the limited access to modern veterinary services. The use of *Acacia* species (including those reclassified as *Vachellia* species) in ethnoveterinary medicine has garnered increasing interest due to their high protein content and medicinal (including anti-parasitic) properties, offering a sustainable source of fodder particularly in arid and semi-arid regions. However, scientific assessment of their efficacy and safety remains limited. This systematic review examines the ethnoveterinary uses, biological efficacy and safety of *Acacia* species across Africa. A literature search was conducted using PubMed, Google Scholar and Scopus, yielding 519 relevant studies published between 2001 and 2024. After applying the inclusion and exclusion criteria, 43 eligible studies were analyzed based on their relevance, geographical location and livestock disease applications. Plants of the World online database was used to validate the names of the species and authority. Ethiopia had the highest usage of *Acacia* species (25%), then Nigeria (20%) followed by both South Africa (15%) and Namibia (15%). *Vachellia nilotica* (*Acacia nilotica*) was the most frequently cited species (26.3%), followed by *Vachellia karroo* (*Acacia karroo*) (15.8%). Ethnobotanical records indicate that the different *Acacia* species have been traditionally used to treat conditions such as diarrhea, wound infections and complications such as retained placenta. Pharmacological studies corroborate the therapeutic benefits of *Acacia* species with evidence of their antimicrobial, anti-inflammatory, antioxidant and anthelmintic effects, though some toxicity concerns exist at high dosages. The systematic review revealed the efficacy and safety (to some extent) of *Acacia* species in livestock disease management, emphasizing their potential integration into veterinary medicine. However, the dearth of in vivo studies underscores the need for pre-clinical and clinical trials to establish safe and effective dosages for use in livestock.

## 1. Introduction

The African continent has a diverse array of livestock breeds that are uniquely adapted to various ecological zones and environmental conditions [1]. In many African countries, livestock farming remains a vital source of income for smallholder and commercial farmers. Livestock is not only a source of food and income but also provides draught power, manure for crop production, employment opportunities and is integral to traditional ceremonies [1,2,3]. As a result, livestock contributes significantly to the socio-economic well-being of many rural households [4]. However, climate change remains a significant factor influencing the health of animals as livestock travel long distances to access communal resources, leading to the mixing of different herds and close contact among animals. This dramatically enhances pathogen transmission and contributes to the spread of diseases across regions [5]. Furthermore, small-scale farmers in rural areas find it challenging to manage the health of their livestock due to limited modern veterinary services, which are often expensive and at times inaccessible [2]. These concerns worsen the impact of prevalent livestock diseases, which can severely affect productivity and threaten livelihoods. Subsequently, many African livestock producers still rely on traditional medicine to treat and manage animal diseases [6].

Ethnoveterinary medicine involves the use of plants and associated indigenous knowledge to treat livestock health problems by communities [7,8,9,10,11]. These practices have evolved over time and continue to be effective and culturally acceptable, especially in regions with limited conventional veterinary care. Among the diverse flora utilized in ethnoveterinary medicine, *Acacia* species are notable for their widespread use in many African countries [12]. The genus *Acacia* consists of approximately 1500 species [13] and is indigenous to various regions around the globe including Africa, Australia and Asia. Following taxonomic revisions, some *Acacia* species in Africa have now been reclassified as *Vachellia* and *Senegalia* species, while those in Australia remain in the genus *Acacia* [13]. Members of the genus *Vachellia* belong to the Acacieae tribe, Fabaceae family and the subfamily Mimosideae [13,14]. *Acacia* and *Vachellia* species are widely utilized in African traditional medicine with different plant parts including leaves, stem bark, pods, seeds and roots prepared as decoctions, infusion and paste to treat wounds, bacterial and fungal infections [15,16].

Previous studies have established that *Acacia* species extracts exhibit various pharmacological properties such as antimicrobial, antioxidant, antiplasmodial, anti-inflammatory and antiparasitic effects [17,18]. Among the several *Acacia* species studied thus far, *Acacia nilotica* stands out for its remarkable medicinal properties against a range of livestock diseases including bacterial, fungal and parasitic infections [19,20,21,22,23]. In addition, several other *Acacia* species are recognized as sources of valuable natural alternatives to commercial drugs due to their richness in phtyochemicals such as flavonoids, phenolics, saponins, alkaloids, tannins and terpenoids [24]. *Acacia* species extracts and their bioactive compounds are known to exert antibacterial effects as demonstrated against pathogenic microorganisms such as *Staphylococcus aureus*, *Escherichia coli* and *Salmonella typhimurium* [25,26]. Other studies have confirmed the wound healing activity of extracts from *Acacia caesia* [27] and *Acacia catchu* [28].

Given the diverse medicinal properties of *Acacia* species, there is a need for comprehensive appraisal to consolidate their therapeutic potential in managing livestock health conditions. Furthermore, understanding the existing empirical data from in vitro and in vivo studies are essential to ascertain their efficacy and safety for integration into modern veterinary medicine. Thus, the current systematic review entails a critical appraisal of the ethnoveterinary uses, biological activities and safety status of *Acacia* species across Africa in an effort to assess their potential role as a viable option for managing livestock health in Africa.

## 2. Results

Based on PRISMA protocol (Figure 1), 519 articles were generated from three databases (Scopus, Google Scholar and PubMed). Following the removal of duplicates, 503 articles were screened for eligibility based on abstract and titles and after a complete evaluation. Thereafter, 460 studies were excluded due to factors such as non-animal studies and reviews. As a result, 43 studies that met the inclusion criteria were used as the core of the review.

### 2.1. Geographical Occurrrence and Ethnoveterinary Practices

Figure 2 illustrates the geographical distribution of ethnoveterinary reports involving *Acacia*/*Vachellia*/*Senegalia* species across the African countries. Ethiopia had the highest (25%) report on ethnoveterinary practices and was closely followed by Nigeria with 20% (Figure 3). Following the regional trends, Table 1 summarizes the 12 *Acacia*/*Vachellia*/*Senegalia* species identified through a systematic review of the literature from the three databases. These plant species are used to manage various livestock diseases, with their modes of application varying depending on the condition treated. Among the plant species, *Vachellia nilotica* (*Acacia nilotica*) was the most frequently cited (25%), followed by *Vachellia karroo* (*Acacia karroo*) (15%) and *Vachellia tortilis* (*Acacia tortilis*) (10%).

Leaves (27%) were the most used plant part to prepare remedies, followed by bark (23%), seeds (18%) and roots (14), while other parts were rarely mentioned (Figure 4a). The preparation method of the medicinal plant parts varied greatly, and remedies were administered through different routes with topical (50%) and oral (43%) applications being the most highlighted techniques (Table 1). *Acacia*/*Vachellia*/*Senegalia* species were widely used to treat various livestock diseases including respiratory problems, wounds, dystocia, diarrhea and retained placenta (Figure 4b).

### 2.2. Biological Activity and Safety

*Acacia*/*Vachellia*/*Senegalia* species extracts demonstrated potent antimicrobial properties, particularly against Gram-positive bacteria such as *Staphylococcus aureus* (Table 2). Extract activity was classified as noteworthy (minimum inhibitory concentration, MIC ≤ 1 mg/mL), moderate (MIC from 1 to 8 mg/mL) or weak (MIC from 8 to 12.5 mg/mL). The methanolic extracts of *Senegaia burkei* exerted good antimicrobial activity against *Staphylococcus aureus* and *Shigella flexneri*, with MIC values as low as 0.25 mg/mL. Moreover, the methanolic extracts of *Vachellia nilotica* were found to be moderately effective against *Staphylococcus aureus*, with an MIC range of 1.56–3.12 mg/mL. On the other hand, ethanolic extract of *Senegalia polyacantha* showed moderate activity against *Staphylococcus aureus* (MIC = 1.56 mg/mL) and *Escherichia coli* (MIC = 3.13 mg/mL), while the hexane extract from *Acacia mearnsii* demonstrated weak activity (MIC > 8 mg/mL) against *Staphylococcus aureus* and *Escherichia coli*. The choice of extraction solvents influenced the effectiveness of extracts, with most organic extracts demonstrating greater activity against tested pathogens compared to the aqueous extracts [39] (Eloff, 1998).

Evidence from different bioassays indicate that *Acacia* species exhibit antioxidant, anti-inflammatory, anthelmintic and cytotoxicity activity (Table 3). Acetone extract of *Vachellia gerrardii* demonstrated strong anti-inflammatory activity with an inhibitory concentration (IC_50_) value of 0.39 mg/mL in the 15-lipoxygenases (15-LOX) inhibition test. The ethanol extract of *Acacia tortilis* also demonstrated inhibitory effects on both cyclooxygenase (COX-1 and COX-2) enzymes. *Senegalia ataxacantha* proved to be an effective antioxidant with its 2,2-diphenyl-1-picrylhydrazyl (DPPH) radical scavenging activity reaching 92.62%. *Senegalia senegal* methanol extract was an effective anthelmintic therapeutic agent at concentrations as low as 0.5 mg/mL within 12 h. Tests on mammalian cells revealed that *Vachellia nilotica* extracts were toxic, with the bark extract having a lethal concentration (LC_50_) value of 0.0029 mg/mL.

### 2.3. Phytochemical Profiles

A total of 24 compounds were isolated from various parts of *Acacia*/*Vachellia*/*Senegalia* species (Table 4). These compounds exhibited antibacterial, antioxidant and antiplasmodial activities, highlighting their potential role in improving animal health.

## 3. Discussion

### 3.1. Ethnoveterinary Practices Associated with Acacia/Vachellia/Senegalia Species in Africa

In African ethnoveterinary practice, pastoralists use local plants to manage livestock health, with *Acacia* species being one of the widely used medicinal plants. The majority of the *Acacia* species reported herein have been documented for their therapeutic benefits in Africa and globally for different conditions [72,73,74,75]. Ethiopia accounted for the highest number of practices using *Acacia*/*Vachellia*/*Senegalia* species in ethnoveterinary medicine, especially for conditions such as dystocia and retained placenta (Table 1). This suggests a strong cultural dependence on these species in regions with limited access to veterinary services [76].

The genus *Acacia* is a multipurpose medicinal plant widely used in traditional medicine practices by indigenous people worldwide to manage human and livestock diseases. The use of *Acacia*/*Vachellia*/*Senegalia* species for treating livestock diseases involves various plant parts, with leaves being the most utilized plant part (Table 1). The high usage of leaves is likely due to their abundance and availability, making it more sustainable than root harvesting [77]. The limited use of roots is particularly beneficial, given that *Acacia*/*Vachellia*/*Senegalia* species are shrubs and trees; extensive harvesting of the roots could lead to environmental disruption and threaten plant sustainability. Also, overharvesting fruit and seeds can negatively impact plant reproduction and overall population sustainability. However, the use of seeds (soaked or boiled) is a common practice in Nigeria, relative to other African countries [35]. Frequency of plant species utilization revealed that *Vachellia nilotica* was the most used plant species which was followed by *Vachellia karroo*. These two plant species are mainly used in Nigeria (*Vachellia nilotica*) and South Africa together with Namibia (*Vachellia karroo*). Moreover, the species are typically utilized in the treatment of cattle, for problems ranging from foot and mouth disease, diarrhea (*Vachellia nilotica*, *Vachellia karroo*), fractures, retained placenta, and bacterial infections.

Plant preparation methods varied depending on the disease condition and the region, with decoction, infusion and maceration being the most common techniques. The application of these techniques is supported by ethnomedicinal studies worldwide [78,79,80,81]. Compared to alternative routes, oral and topical applications for internal and external conditions are mostly preferred. The application of these techniques in traditional medicine is due to their minimal equipment requirement and simplicity especially in rural and low-resource regions.

### 3.2. Antimicrobial Activity of Acacia/Vachellia/Senegalia Species

Antimicrobial properties of plants remain of interest in the search for new antibiotics and fungicidal drugs [82]. As shown in this review, the antimicrobial properties of the extracts of the *Acacia* plants against various bacterial and fungal strains are well recognized in vitro, while in vivo studies are limited (Table 2). Experimental work on *Acacia* frequently reports on bacterial strains such as Gram-negative *Escherichia coli* and *Pseudomonas aeruginosa*, and Gram-positive *Staphylococcus aureus* and *Bacillus subtilis*. Furthermore, *Candida* species were the most studied fungal strains. Antimicrobial properties of *Senegalia catechu* methanol extracts demonstrated good activity with minimum inhibitory concentration (MIC) ranging from 0.7 to 2 mg/mL against *Bacillus subtilis*, *Staphylococcus aureus*, *Salmonella typhi*, *Escherichia coli*, *Pseudomonas aeruginosa* and *Candida albicans* strains [43]. The observed activity was primarily due to the phytochemicals extracted more effectively by the organic solvent than by water. *Vachellia sieberiana* leaf extract displayed noteworthy activity against resistant clinical isolate *Staphylococcus aureus* with an MIC value of 0.032 mg/mL [83]. The extract showed moderate activity against other resistant clinical isolates, demonstrating an activity spectrum against 87.5% of the bacteria tested. The work by Kirabo et al. [84] corroborated the efficacy of *Vachellia sieberiana* extracts with MIC of 0.16 mg/mL when tested against *Salmonella paratyphi* and *Pseudomonas aeruginosa* strains. *Vachellia nilotica* bark extracts (acetone and water) were effective (0.039–0.3130 mg/mL) against several clinical isolates, ATCC and field strains, while the leaf extracts (acetone and water) showed good to moderate inhibitory activity [48]. On the contrary, *Vachellia nilotica* fruit extract (methanol) showed weak activity at concentrations ranging from 12.5 to 100 mg/mL against *Staphylococcus aureus*, *Bacillus subtilis*, *Pseudomonas aeruginosa* and *Escherichia coli* [85]. Fractionation of *Mimosa decurrens* stem bark using ethyl acetate and methanol revealed inhibitory activity (0.0125 mg/mL) comparable to the standard antibiotic ampicillin [40]. In another study by Fatimah [86], the aqueous extracts of *Acacia senegal* and *Vachellia tortilis* varied in their efficacy in inhibiting the mycelial growth of tested fungal isolates. Specifically, *Vachellia tortilis* exhibited moderate inhibition against *Fusarium solani* and *Helminthosporium ostratum* at 5% concentration, while *Acacia senegal* showed weak inhibition of *Fusarium solani* and no activity against *Helminthosporium rostratum*. The triterpenoid compounds (20S-oxolupane-30-al, 20R-oxolupane-30-al, and betulinic acid) isolated from the stem bark of *Senegalia mellifera* were found to have high inhibitory activity against *Staphylococcus aureus*, while no activity was observed against *Escherichia coli* and *Enterococcus faecialis* [87]. In a study by Thanish Ahamed et al. [88], the authors evaluated the antimicrobial activity of taxifolin, a flavonoid isolated from *Senegalia catechu* leaf extract. They found that the compound exhibited moderate antibacterial activity against *Streptococcus mutans* and *Lactobacillus acidophilus* when compared to the reference drug, chlorhexidine. The application of silver nanoparticles from *Vachellia xanthophloea* extracts showed excellent activity against *Staphylococcus aureus* (MIC = 0.04 mg/mL, MBC = 0.04 mg/mL) and good activity for *Escherichia coli* (MIC = 0.33 mg/mL, MBC = 0.66 mg/mL) [51].

The potency of compounds against various pathogens could be due to their ability to interact with microbial membrane proteins. This interaction can lead to alterations in cell wall integrity, enzymatic activity and inactivation of key proteins within the microorganisms [66,89,90,91]. These findings suggest that *Vachellia* extracts could be developed as a natural alternative to conventional antibiotics especially in areas where antibiotic resistance is of concern. However, there is still a need for research on the underlying mechanism by which *Vachellia* extracts exert their antimicrobial effects. Studies that explore whether the plant compounds interfere with cell membranes or metabolic pathways could provide valuable insight. Another critical area to address is the scarcity of in vivo studies. Future research should prioritize animal trials to substantiate the efficacy of the plant extracts and compounds, thereby bridging the gap between promising in vitro results and practical application. Finally, exploring the potential for synergistic effects between *Acacia* extracts and conventional antibiotics could open new avenues for overcoming antimicrobial resistance and enhancing treatment outcomes.

### 3.3. Antiparasitic Activity

Several medicinal plants with anthelmintic potential have been studied in vitro and/or in vivo for their activity in various life stages of parasites including eggs, larvae and adult worms. Building on this evidence, in vitro studies have evaluated the anthelmintic properties of *Acacia* species such as *Vachellia nilotica* and *Vachellia tortilis*, demonstrating significant dose-dependent activity against *Haemonchus contortus*. For instance, an egg hatching inhibition assay (EHT) demonstrated that *Vachellia tortilis* was effective against *Haemonchus contortus* and *Caenorhabditis elegans* with an inhibitory concentration (IC_50_) of 1.58 mg/mL for the aqueous extract and 0.58 mg/mL for the acetone extract, while aqueous extract of *Vachellia nilotica* had more larvicidal activity with an IC_50_ of 0.195 mg/mL [92]. Similarly, Badar et al. [19] found that exposing *Haemonchus contortus* eggs to crude methanol extract of *Vachellia nilotica* resulted in a dose-dependent inhibition of egg hatching, with an LC_50_ of 0.2 mg/mL for bark and 0.77 mg/mL for leaves. *Vachellia tortilis* acetone extract showed 100% larval mortality at 2.5 mg/mL and a lethal concentration (LC_50_) of 0.84 mg/mL against *Haemonchus contortus* [92]. The work by Zarza-Albarran et al. [93] further confirmed that bioactive compounds from *Vachellia farnesiana* pods possess potent anthelmintic activity against *Haemonchus contortus* eggs. Other studies have also highlighted that the antiparasitic effect of plants vary depending on the plant species type, the targeted nematode and the parasite stage. For instance, Moreno-Gonzalo et al. [94] found that the phenolic extracts from Ericaceae were more active against eggs and third-stage larvae (L_3_) of *Teladorsagia circumcincta* compared to its activity against *Haemonchus contortus* and *Teladorsagia circumcincta*. Moreover, *Agave sisalana* and *Moringa oleifera* extracts showed greater ovicidal effect whereas *Vachellia nilotica* and *Vachellia tortilis* were more active against L_3_ of gastrointestinal nematodes (GIN) [92,95,96]. These differences are likely due to the variation in the enzymatic components and membrane structures of the nematode species and life stages.

In vivo studies also support the antiparasitic properties of the *Acacia* species. Castillo-Mitre et al. [97] confirmed the in vivo anthelmintic activity of *Vachellia cochliacantha* leaves against *Haemonchus contortus* in Boer goat kids. The study showed that the mixture of goat feed with 5% of *Vachellia cochliacantha* significantly reduces the fecal egg count in Boer goat kids, suggesting an alternative treatment for nematodiasis. Similarly, in sheep, crude powder and methanolic extract of *Vachellia nilotica* bark and leaves exhibited dose-dependent anthelmintic activity [19]. The study recorded a maximum reduction of 72.01% in fecal egg counts for the methanolic extract of bark and 63.44% for leaves. The anthelmintic efficacy of medicinal plants is largely attributed to the composition of their phytochemical constituents. Minho et al. [98] confirmed that drenching sheep with condensed tannin extract (CTE) of *Acacia molissima* reduced fecal egg counts and worm burden in the abomasum. The effect of tannins may be due to their affinity to bind parasite proteins causing structural damage to the cuticle [99]. Additionally, tannins have been shown to interfere with oxidative phosphorylation and blocking ATP synthesis in *Haemonchus contortus* [100]. Based on these findings, the extensive use of *Acacia* for parasites in ethnoveterinary medicine can be substantiated by the in vitro and in vivo studies which show promising potential in preventing and treating GINs. However, many of the plant species remain unproven for their safety or toxicity. This may lead to limited knowledge of their potential adverse effects and difficulties in the identification of the safe and most efficient treatments. Future research should focus on exploring the synergistic or combined effect of various plant extracts against parasitic infections.

### 3.4. Antioxidant Activity

The utilization of plants is associated with a reduced risk of diseases in livestock such as metabolic disorders, immune system dysfunction and inflammatory conditions. Plant bioactive compounds have health-promoting effects that possess antioxidant properties which help protect against cellular damage and promote overall animal health. Antioxidants play a significant role in neutralizing free radicals thereby reducing oxidative stress [101,102,103]. Oxidative stress is a common underlying factor in dysfunctional immune and inflammatory responses, increasing an animal’s susceptibility to numerous health disorders [104,105,106]. The antioxidant potency of *Acacia* extracts has been confirmed in various assays using 2,2-diphenyl-1-picrylhydrazyl (DPPH), 2,2-azino-bis(3-ethylbenzothiazoline-6-sulfonic acid (ABTS) and ferric reducing antioxidant power (FRAP), demonstrating their ability to scavenge free radicals and reduce oxidative damage [56,107,108]. A study by Delgadillo Puga et al. [109] evaluated the antioxidant activity and protection against oxidative-induced damage of extracts from *Acacia shaffneri* and *Vachellia farnesiana* pods. The study found that pod extracts effectively reduced the harmful effects of free radicals as demonstrated by their strong scavenging activity against ABTS^+^ and DPPH^+^. The free radical scavenging capacity of ethyl acetate soluble fraction of *Vachellia farnesiana* was calculated at an IC_50_ value of 0.0215 mg/mL compared to that exhibited by the standard butylated hydroxytoluene (BHT) which had an IC_50_ value of 0.0204 mg/mL [110]. The antioxidative potential of *Vachellia farnesiana* was attributed to the presence of gallic acid in the extract. In a DPPH assay, the methanolic extract of *Vachellia pennatula* pods had a significantly higher antioxidant activity than the gallic acid and Trolox, which are standard controls used when determining the scavenging potential of extracts [111].

A study by Sowndhararajan et al. [108] found that acetone and methanol extracts from the bark of *Acacia* species exhibited good antioxidant activity, which was attributed to the high phenolic and flavonoid content in the extracts. Similarly, acetone extracts of *Vachellia nilotica* leaves demonstrated strong antioxidant activity when using different antioxidant test systems, which is attributed to the presence of tannins in the extracts [112]. Therefore, the antioxidant activity of the *Acacia* species could be justified by the synergistic action of the secondary compounds contained in the extract, which are known to donate electrons to neutralize reactive oxygen species (ROS). According to the work by [113], using different extraction methods reveals that differences in solvents significantly influenced the ABTS radical scavenging activities of *Acacia* samples. In line with the work, the methanol extract of *Vachellia nilotica* leaves exhibited the highest antioxidant activity at 94.3% followed by the ethyl acetate extract, which showed 90.7% activity [114]. Furthermore, at a concentration of 0.285–0.455 mg/mL the leaves of *Vachellia nilotica* were found to have a significantly higher antioxidant activity compared to the pods and bark [115]. The results on the antioxidant potential of *Acacia* species are comparable to that of other medicinal plants traditionally used to treat livestock diseases. For example, *Moringa oleifera* [116,117] and *Azadirachta indica* [118,119] are known for their high antioxidant content and studies have shown that they exhibit effective radical scavenging activities to those found in *Acacia* species. These findings provide evidence of the ability of *Acacia* species to neutralize reactive oxygen species which are known to resist oxidative damage and combat diseases related to oxidative stress [120]. Therefore, incorporating *Acacia* extracts into livestock diets could offer a natural and effective way to enhance antioxidant defence, potentially reducing the incidence of oxidative stress-related diseases and improving overall animal health and productivity. However, the efficacy of the extracts may vary depending on factors such as the plant species, plant part, age of plant, chemotype, extraction solvent [121,122] and the method applied [123]. Therefore, future studies should focus on conducting more in vivo trials to evaluate the antioxidant effects of *Acacia* extracts on health, immune function and productivity.

### 3.5. Anti-Inflammatory Activity

The anti-inflammatory potential of medicinal plants has been a subject of interest because they have been used in ethnomedicine and ethnoveterinary medicine to treat inflammatory ailments. As chronic inflammation is the root of many diseases, research continues to search for safe and effective anti-inflammatory agents in both human and veterinary medicine [124]. Medicinal plants are thought to exert their anti-inflammatory effects through mechanisms such as inhibition of pro-inflammatory cytokines, 15-lipoxygenases (LOX), nitric oxide synthase (NOS) and cyclooxygenase (COX) enzymes [125].

In this review, some *Acacia* species showed significant anti-inflammatory activity, especially by inhibiting key enzymes such as COX-1, COX-2 and 15-LOX. For instance, the acetone extract of *Vachellia gerrardii* showed potent inhibitory activity against 15-LOX with an IC_50_ of 0.39 mg/mL, compared to the IC_50_ of 35 mg/mL for quercetin, which could make it a possible natural anti-inflammatory agent [54]. In an in vitro study, leaf extracts and fractions of *Acacia mearnsii* exerted anti-inflammatory effects by inhibiting the production of pro-inflammatory cytokines enzymes in LPS-stimulated RAW 264.7 macrophage cells, specifically through the suppression of IL-1β, COX-2 and NO production [126]. The analyzed fractions showed variation in their inhibitory mechanism; some were more potent towards the release of NO while some inhibited the mRNA expression levels of the anti-inflammatory cytokine IL-1β, COX-2, iNOS and IL-6, which might be attributed to the total phenolic content (646.6 mg/g) and primarily proanthocyanidins (12.6 mg/g).

The anti-inflammatory activity of *Vachellia farnesiana*, *Vachellia tortilis* and *Acacia longifolia* when studied using COX-1 and COX-2 enzymes demonstrated good inhibitory activity when compared to the positive controls, indomethacin and NS-398 [55]. According to the authors, variation in total phenolic composition in species showed a strong inverse correlation to the COX-2 inhibitory activity. Another in vitro study revealed that the ethanolic extract of *Senegalia caesia* inhibited protein denaturation with the highest concentration (0.4 mg/mL) achieving 99% inhibition [127]. Fractionation of *Senegalia polyphylla* (BH) contributed to the high PGE_2_ inhibition percentage of 74.8% which was better than the reference drugs dexamethasone and indomethacin [128]. The excellent anti-inflammatory activity in *Senegalia polyphylla* slightly declined when the isolated glycosylated flavonoid, astragalin, was tested revealing a moderate inhibition of prostaglandin (PGE_2_) at 48.3% which was statistically different from the negative control, dexamethasone and indomethacin [128]. *Senegallia polyphylla* extract fraction was more effective, and more practical than the isolated compound (astragalin), owing to the synergistic interactions among multiple phytochemicals.

In vivo investigations using animal models have also provided convincing evidence for the anti-inflammatory properties of *Acacia* species. For instance, dichloromethane stem bark extract in *Senegalia mellifera* revealed a dose-dependent anti-inflammatory effect by reducing the inflamed paw diameter in mice of both carrageenan- and formalin-induced paw edema models [129]. The extract was effective in inhibiting pain sensation through the peripheral and central mechanisms. The anti-inflammatory activity of *Senegalia mellifera* can be attributed to the concoction of phytochemicals in the plant that contributed to its analgesic and antinociceptive potential. Similarly, the ethanolic *Senegalia caesia* extract at a high dose (0.4 mg/mL) showed anti-inflammatory activity by suppressing paw swelling with 95.30% inhibition in comparison to the standard drug indomethacin [127]. Evidence from the studies indicates that *Senegalia caesia* is effective in managing acute inflammatory conditions.

In a formalin-induced paw edema model, mice treated with the aqueous bark extract of *Vachellia nilotica* at a dose of 150 mg/kg body weight exhibited 57.16% inhibition of inflammation which was comparable to the standard drug diclofenac, which showed 58.10% inhibition [130]. Furthermore, the administration of *Vachellia nilotica* methanolic seed extract at concentrations ranging from 50 to 200 mg/kg demonstrated a significant anti-inflammatory effect in a carrageenan-induced rat paw edema assay comparable to that of diclofenac and also showed a dose-dependent (0.5–2.5 mg/mL) inhibitory effect [131]. The authors also demonstrated the anti-inflammatory activity of the methanolic extract at a concentration of 200 mg/kg by inhibiting COX-2 enzyme activity. The summarized findings reveal that certain *Acacia* species possess strong anti-inflammatory properties with IC_50_ concentrations comparable to non-steroidal inflammatory drugs (NSAIDs). While the results confirm the ethnopharmacological use of the *Acacia* plants, further studies are needed to isolate bioactive compounds responsible for the confirmed activity and to evaluate their efficacy and safety in in vivo studies.

### 3.6. Toxicity Evaluation of Acacia/Vachellia/Senegalia Species

Traditional medicinal plants are considered potential sources of unique phytochemicals with promising pharmacological activities. Assessing the toxicological profile of these phytochemicals is essential to ensure their safe therapeutic application in ethnoveterinary medicine. Currently, several studies have investigated the toxicity of various *Acacia* species (Table 3). For example, Sserunkuma et al. [48] highlighted that acetone and water extracts of *Vachellia nilotica* bark were cytotoxic, with LC_50_ values of 0.0332 and 0.0278 mg/mL, respectively. The authors reported that *Vachellia nilotica* exhibited high cytotoxicity, as indicated by a selectivity index (SI) of less than 1 for its antibacterial activity. This suggests that the observed antibacterial activity of *Vachellia nilotica* may primarily result from general cytotoxicity rather than specific antimicrobial properties. Ethanol seed extract from *Senegalia catechu* was observed to be cytotoxic in hepatocellular carcinoma (HepG2) cells, with an IC_50_ of 0.077 mg/mL [132]. The observed cytotoxicity of *Senagalia catechu* extract, mediated primarily through apoptosis, suggests potential application for anticancer bioassays. However, the lack of selectivity could pose risks to normal cells, highlighting the need for further studies to assess its therapeutic index and safety levels. In contrast, other studies have reported on the safety of certain *Acacia* extracts, highlighting the plant’s non-toxic nature. Mattana et al. [133] evaluated the cytotoxicity and genotoxicity of hot aqueous and ethanol extracts of *Vachellia aroma* using Vero cells, showing that both extracts were safe at the cellular and genomic level at concentrations below 5 mg/mL. Furthermore, *Acacia* species demonstrated low cytotoxic effects and minimal genotoxicity on human erythrocyte [134] and colon cells [135].

In animal models, *Senegalia ataxacacntha* methanol extract exhibited no toxicity with an oral lethal dose (LD_50_) greater than 5000 mg/kg body weight [67]. The study also reported that extract concentration at 400 mg/kg increased different parameters in the liver (alanine transaminase, aspartate transaminase, alkaline phosphatase) and kidney (creatinine, urea and sodium ion). However, there were observable signs of moderate glomerular necrosis and lymphocytes hyperplasia on the kidney, hepatocellular necrosis in the liver, and mild mucosa necrosis on the stomach tissues from extract of varying concentrations such as 50, 200 and 400 mg/kg. This suggests that prolonged use of the *Senegalia ataxacacntha* at high concentrations has potential to harm the liver, kidney and stomach. When *Vachellia nilotica* leaves (methanol extracts) were tested in vivo, no signs of toxicity were observed in rats [136]. In addition, Adewale [137] found that aqueous root extracts of *Vachellia nilotica* at 500 mg/kg body weight were not detrimental on the basis of the assessed hematological parameters or deaths caused over 28 days. A similar trend was also reported by Umaru et al. [138] who observed no signs of toxicity in animal models that were given aqueous extract of *Vachellia nilotica* (pods), but the animal models showed a significant increase in body weight. Sarkiyayi et al. [139] explored the effect of ethanolic leaf extract of *Vachellia nilotica* in phenylhydrazine-induced anemia in rats. Findings indicated an increase in the tested blood parameters in the treated group, showed no change in liver biomarkers, suggesting that the ethanolic leaf extract at 100 mg/kg and 200 mg/kg is safe, and can be used in the management of anemia. Likewise, the work by Thangavelu et al. [140] demonstrated that the daily administration of *Senegalia catechu* (seeds) ethanolic extract at doses ranging from 250 to 1000 mg/kg did not cause any mortality in rats or any significant changes in body weight, the relative weight of vital organs and hematological parameters. Lastly, Magnini et al. [141] found that hydroethanolic extract of *Senegalia senegal* leaves was non-toxic in a subacute study, with no deaths or clinical signs of toxicity observed at doses of up to 1000 mg/kg/day over 28 days. While certain *Acacia* extracts, especially those from *Senegalia catechu* and *Vachellia nilotica*, may display cytotoxic effects in vitro, the safety of others has been demonstrated in animal models supporting their potential use in medicinal application. Further research should prioritize more in vivo studies to establish safety profiles of various *Acacia* extracts in a wider range of animal models and dosages. While existing evidence indicates that many *Acacia* extracts such as those from *Vachellia nilotica*, *Senegalia catechu* and *Senegallia senegal* exhibit low toxicity and may even improve certain health parameters, additional studies should focus on long-term safety assessment and potential sub-chronic and chronic effects.

These findings emphasize the importance of evaluating both the safety and efficacy of *Acacia* extracts before recommending them for widespread ethnoveterinary application. A further challenge relates to differences between dosages used in traditional systems and those tested under laboratory conditions. In traditional practice, remedies are commonly prepared as decoctions or infusions and administered in approximate household measures with dosages rarely specified or adjusted according to the livestock species and size (Table 1). These practices are generally considered safe by livestock farmers based on observation. In contrast, experimental studies use standardized extract concentrations expressed in mg/kg which may be higher than the traditional doses. These differences complicate the translation of laboratory results into practical veterinary use and highlight the need for bridging studies that test traditionally relevant preparations under controlled in vivo conditions.

### 3.7. Phytochemicals in Acacia/Vachellia/Senegalia Species

Woody plants are known to serve as rich sources of pharmacologically active secondary metabolites [142,143]. The bioactive compounds identified in the eligible studies were characterized using advanced analytical techniques such as high-performance liquid chromatography (HPLC), column chromatography, mass spectrometry (MS), nuclear magnetic resonance (NMR) spectroscopy as well as basic methods including thin-layer chromatography (TLC) and ultraviolet–visible (UV-Vis) spectrophotometry. These techniques provide reliable metabolite profiling, structural elucidation and compound quantification [144], which are essential for validating bioactivity findings and ensuring reproducibility in phytochemical studies [145].

This review identified 24 bioactive compounds extracted from different plant parts (leaves, stem, bark) of four *Acacia* species, namely *Senegalia ataxacantha*, *Senegalia polyacantha*, *Vachellia xanthophloea* and *Acacia saligna* (Table 4). Phytochemical analysis revealed that *Acacia* species are rich in phenolic compounds, especially flavonoids which have been profiled in the plants. Other major classes of compounds included tannins, saponins, alkaloids, terpenoids, sterols and polysaccharides, which also contribute to the pharmacological potential of the species.

Among the various phytochemicals reported, flavonoids, especially quercetin and epicatechin, were the frequently identified compounds and are well-known for their biological properties [146,147]. These compounds have demonstrated significant in vitro effects, such as antibacterial, anti-inflammatory and antioxidant properties. For example, Ashu et al. [69] reported the antibacterial efficacy of 10 compounds isolated from *Vachellia polyacantha* against 14 multidrug-resistant *Staphylococcus* strains. Epicatechin was reported to be effective in prolonging the lag phase of the bacterial growth kinetics and inhibit proton-ATPase pump activity in the studied bacterial strains. Betulinic acid-3-trans-caffeate, a triterpenoid isolated from *Senegalia ataxacantha*, was found to be active against the different tested strains especially *Staphylococcus epidermidis* and *Candida albicans* with MIC values of 0.0125 mg/mL [17]. The biological activity of compounds isolated from *Acacia saligna* varied depending on the specific bioactive compound, with spirostane saponin exhibiting good cytotoxicity when tested against the HEPG2 (liver cancer) cell line, while biflavonoid glycoside demonstrated strong antioxidant activity [68].

Bioactive compounds in *Acacia* species such as β-sitosterol, epigallocatechin, flavan-3-olsgalloyl and phentolamine have diverse biological activities including antilisterial, anti-inflammatory, antihypertensive, antispasmodic, antimicrobial and cytotoxic effects [148]. Terpenoids are well known for their anti-inflammatory properties, often acting through the inhibition of COX enzymes and suppression of pro-inflammatory cytokines [149]. Additionally, sterols, particularly β-sitosterol, have demonstrated both antioxidant and immunological effects. A study by Cheng et al. [150] revealed that dietary β-sitosterol supplementation regulated cholesterol levels and improved immune function and intestinal oxidative status in broilers.

The therapeutic potential of *Acacia*/*Vachellia*/*Senegalia* species can be linked to the mechanism of their phytochemicals. Flavonoids and phenolics exert antioxidant effects by scavenging reactive oxygen and nitrogen species, chelating transition metals and activating endogenous defence pathways including Keap1-Nrf2-ARE [151]. These actions help protect against oxidative stress-related tissue damage that underlies inflammatory and metabolic disorders in livestock. Similarly, tannins, flavonoids and saponins display antibacterial activity through protein precipitation, disruption of microbial membranes, inhibition of enzyme activity and interference with quorum sensing [151,152,153]. In addition, certain flavonoids and alkaloids inhibit bacterial efflux pumps, interfere with nucleic acid synthesis and disrupt coenzyme-dependent metabolic pathways [69,154]. Together, these mechanisms suppress pathogens such as *Escherichia coli*, *Staphylococcus aureus* and *Candida albicans* which are major causes of infectious diseases in livestock [153,155]. These mechanisms highlight how plants contribute to the prevention and management of oxidative stress and infectious diseases in livestock, supporting their ethnoveterinary relevance.

### 3.8. Limitations

Most of the available studies reported in vitro assays rather than in vivo or clinical trials, and there were inconsistencies in the reporting of dosages and phytochemical data. These limitations highlight the need for standardized dosages and comprehensive in vivo studies in future research. The review process was limited as a single reviewer conducted it, and only English-language studies were included.

## 4. Materials and Methods

### 4.1. Search Strategy

The literature on the genus *Acacia* in African livestock disease management was compiled from three databases including Scopus, PubMed and Google Scholar. The systematic review search was conducted in January 2025 and includes research articles published from 2001 to 2024, reflecting over two decades of relevant scientific literature. The search was conducted using the following keywords: ethnoveterinary OR ethnomedicinal AND biological activity AND phytochemistry OR chemical compounds AND livestock diseases AND *Acacia* OR *Vachellia* AND toxicology AND Africa. Boolean operators and exact search strings were used, such as the following examples: (“*Acacia*” OR “*Vachellia*” OR “*Senegallia*”) AND (“ethnoveterinary” Or “ethnomedicinal”) AND (“livestock” OR “animal health”) AND (“toxicology” OR “pharmacology” OR “biological activity”). In total, the initial search yielded 519 articles from the three databases. After a screening and eligibility check, 43 research articles met the inclusion and exclusion criteria and were included in the review (Figure 1). Articles were initially screened based on English language, title and abstract. The full-length papers were obtained and evaluated if abstracts of the papers met the inclusion criteria. All keywords were searched electronically, while the titles and abstracts were screened manually. This review was conducted in accordance with the PRISMA 2020 guidelines for reporting systematic reviews. No protocol was registered for this review.

### 4.2. Inclusion/Exclusion Criteria and Data Extraction

Research articles were included if they described the use of *Acacia* species in livestock disease management and met the following inclusion criteria: (a) in vitro/in vivo efficacy/safety of *Acacia* species against livestock diseases, (b) biological properties, (c) original research articles and (d) written in English (Figure 1). Exclusion criteria included non-African studies, non-livestock related research, non-English articles and reviews and papers lacking clear species identification. Given the taxonomic reclassification of *Acacia* into *Vachellia* and *Senegalia*, all species names reported in older studies were cross-checked and standardized using the Plants of the World Online (POWO). Where older studies used names from the former *Acacia* classification, the original name reported in the article and the currently accepted synonym were indicated in this review to maintain consistency. No studies were excluded based on outdated nomenclature provided the identification of the plant species is reliable and confirmed. Data were extracted from eligible articles using a standardized form. The following information was included: plant name, country name, extracts, route of administration, type of livestock, target diseases, dosage or concentration, strain tested, type of bioassay (in vitro/in vivo), potential toxicity and phytochemicals. Biological activities including antimicrobial, antioxidant, anti-inflammatory and antiparasitic effects were recorded along with effective and cytotoxic concentrations and references. As this review followed a narrative synthesis approach, no quantitative effect measures were calculated. Outcomes were interpreted descriptively based on the reported findings.

### 4.3. Quality Assessment

This review did not apply any formal risk of bias assessment tools. Potential bias was considered informally based on the narrative nature of the synthesis and diversity of included studies.

## 5. Conclusions

Based on the current systematic review, *Acacia* species, especially *Vachellia nilotica*, have been effectively used to treat several livestock diseases due to their rich bioactive compounds with proven antimicrobial, anti-inflammatory, antioxidant and antiparasitic properties. These medicinal properties highlight the potential of *Acacia* as a promising, safer and natural alternative to commercial drugs for managing livestock health. To complement their pharmacology promise, it is crucial to recognize and build upon the rich indigenous knowledge that have long guided their traditional use in livestock health care. However, this knowledge remains underexplored in research and product development contexts. To fully realize both the pharmacological and economic potential of *Acacia*, future research should integrate ethical bioprospecting approaches that combine indigenous knowledge with robust scientific validation. This includes comprehensive in vitro, in vivo and clinical studies, as well as efforts to develop culturally appropriate, standardized and accessible veterinary solutions. Priority should be given to in vivo validation of *Acacia* species for livestock diseases such as helminthiasis, mastitis and diarrhea where traditional use is widespread but experimental evidence is scarce. Research should also expand beyond cattle to under-studied livestock such as goats and poultry. In addition, the need for phytochemical profiling and elucidation of the mechanism of action is essential for exploring the ethnoveterinary potential of *Acacia* species. Furthermore, ecological sustainability must not be overlooked. Overexploitation of bark and roots from *Acacia* species may compromise the long-term plant survival while unsustainable harvesting of fruits and seeds can hinder natural regeneration. Therefore, promoting the use of leaves and pods, encouraging cultivation and integrating community-based conservation strategies will be essential to ensure that wider ethnoveterinary use does not threaten biodiversity.

## Figures and Tables

**Figure 1 plants-14-03107-f001:**
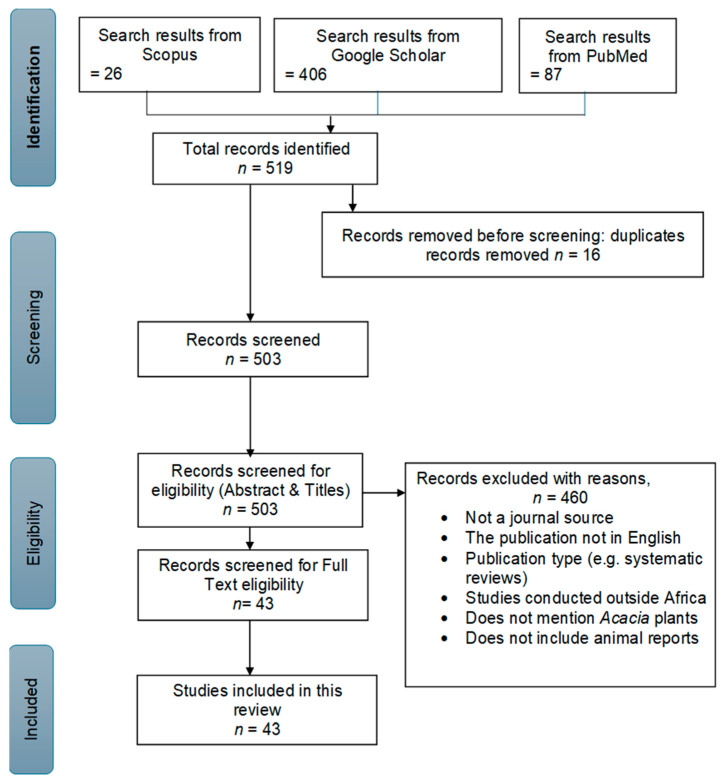
Flow chart of search methodology applied in the systematic review.

**Figure 2 plants-14-03107-f002:**
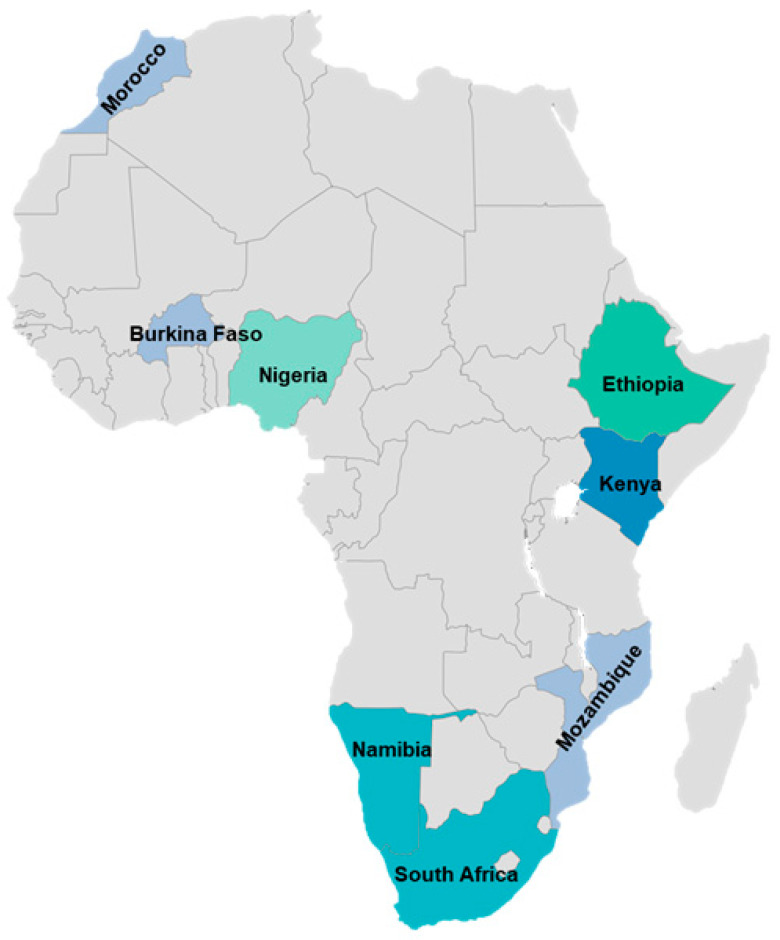
Geographical location of African countries reporting the ethnoveterinary uses of *Acacia*/*Vachellia*/*Senegalia* species.

**Figure 3 plants-14-03107-f003:**
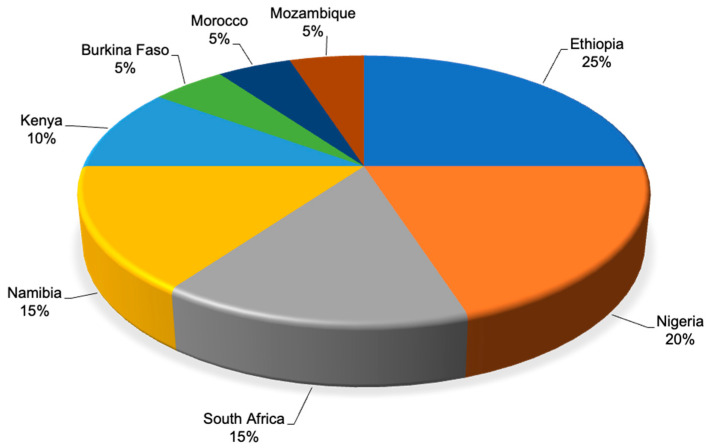
Distribution of studies focusing on the ethnoveterinary uses of *Acacia*/*Vachellia*/*Senegalia* species across African countries.

**Figure 4 plants-14-03107-f004:**
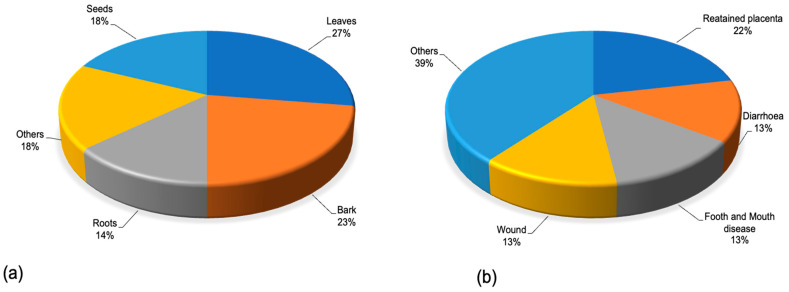
Distribution of (**a**) plant parts used in ethnoveterinary practices and (**b**) livestock ailments treated with *Acacia*/*Vachellia*/*Senegalia* species.

**Table 1 plants-14-03107-t001:** Ethnoveterinary uses of *Acacia species* (including *Vachellia* and *Senegalia* based on recent taxonomic revisions) for managing animal health in Africa.

Plant Species (Accepted Name)	Plant Species (Synonym)	Plant Part Used	Disease/Condition	Livestock	Country	Preparation Method	Dosages and Mode of Application	Reference
*Acacia mearnsii* De Wild	-	Leaves	Coughing	Ruminants, pigs, poultry	Kenya	Infusion	Not specified.	[29]
*Acacia* sp.	-	Leaves	Wound	Small ruminants	Mozambique	Leaves are ground along with salt	Ground material is topically applied.	[16]
*Faidherbia albida* (Delile) A. Chev.		Roots	Urinary problems	Sheep	Morocco	Decoction	Root decoction is administered orally to the diseased animal.	[10]
*Senegalia asak* (Forssk.) Kyal. & Boatwr.	*Acacia asak* (Forssk.) Willd.	Bark	Dystocia and retained placenta	Large and small ruminants	Ethiopia	Bark is used as fresh material	Bark is tied to the neck of the diseased animal.	[15]
*Senegalia macrostachya* Rchb. ex DC.	*Acacia macrostachya* Rchb. ex DC.	Leaves	Anthelmintic	Cattle	Burkina Faso	Maceration	Leaves are given to the cattle orally (two doses) on two consecutive days.	[30]
*Vachellia abyssinica* Benth.	*Vachellia abyssinica* Benth.	Leaves	Snakebite	Bovine	Ethiopia	Not specified	Freshly ground leaf juice is applied topically (daily) until the wound heals.	[31]
*Vachellia drepanolobium* (Harms ex siostedt)	*Acacia drepanolobium* Harms ex siostedt	Bark	Retained placenta	Cattle	Kenya	Decoction	Not specified.	[29]
*Vachellia erioloba* E. Mey	*Acacia erioloba* E. Mey	Bark	Retained placenta	Not specified	Namibia	Infusion	Infusion is given orally to the diseased animal.	[32]
*Vachellia karoo* Hayne	*Acacia karoo* Hayne	Bark	Fractures and Diarrhea	Cattle	South Africa	Not specified	Not specified.	[33]
*Vachellia karoo* Hayne	*Acacia karoo* Hayne	Roots	Eye problems	All livestock	Namibia	Infusion	Infusion is applied topically to treat eye inflammation.	[32]
*Vachellia karroo* Hayne	*Acacia karroo* Hayne	Bulb	Retained placenta and bacterial infection	Cattle	South Africa	Maceration	Bulbs are given orally to the diseased animal.	[34]
*Vachellia nilotica* (L.) Wild	*Acacia nilotica* (L.) Wild	Seed	Diarrhea	Cattle	Nigeria	Seeds are soaked in water for 24 h	The mixture is administered orally to the diseased animals.	[35]
*Vachellia nilotica* (L.) Wild	*Acacia nilotica* (L.) Wild	Bark	Retained placenta	Cattle	Namibia	Infusion	Infusion is administered orally to the diseased animal.	[32]
*Vachellia nilotica* (L.) Wild	*Acacia nilotica* (L.) Wild	Seeds	Foot and mouth disease	Cattle and sheep	Nigeria	Infusion	Not specified.	[36]
*Vachellia nilotica* (L.)	*Acacia nilotica* (L.)	Seeds	Foot and mouth disease	Cattle	Nigeria	Seeds are boiled with salt	Mixture is used to wash the affected areas.	[35]
*Vachellia nilotica* (L.)	*Acacia nilotica* (L.)	Seeds	Foot and mouth disease	Cattle	Nigeria	The seeds are soaked in cow urine for 24 h	Infusion is used to wash the affected areas.	[35]
*Vachellia oerfota* Forssk.	*Acacia oerfota* (Forssk.)	Leaves, stem, roots	Kidney problem	Cattle	Ethiopia	Decoction	Decoction is administered orally (1–2 cups).	[37]
*Vachellia sieberiana* DC.	*Acacia sieberiana* DC.	Leaves	Wound	Large ruminants	Ethiopia	Leaves are crushed and mixed with water	Mixture is applied as a topical ointment to the wounded area.	[15]
*Vachellia tortilis* (Forssk.) Galasso & Banfi	*Acacia tortilis* (Forssk.) Hayne	Branch tips	Diarrhea	Cattle	South Africa	Not specified	Not specified.	[33]
*Vachellia tortilis* (Forssk.) Hayne.	*Acacia tortilis* (Forssk.) Galasso & Banfi	Gum	Wound	All livestock	Ethiopia	Not specified	Gum is applied topically to the wounded area.	[38]

**Table 2 plants-14-03107-t002:** In vitro antimicrobial activity of *Acacia* species.

Plant Species (Accepted Name)	Plant Species (Synonym)	Plant Part	Extraction Solvent	Strain(s) Tested	Extract(s) Concentration	Summary of the Tested Extracts	Reference
*Acacia decurrens* Willd.	*Mimosa decurrens* J.C. Wendl.	Stem bark	Ethyl acetate, methanol	*Staphylococcus aureus*, *Escherichia coli*, *Salmonella typhi*, *Klebsiella pneumoniae*, *Micrococcus luteus*, *Shigella sonnei*, *Staphylococcus epidermis*, *listeria monocytogenes*, *Enterococcus faecalis*	0.0125–0.05 mg/mL	Ethyl acetate and methanol fractions showed excellent activity against the tested microorganisms with MIC value of 0.0125 mg/mL, comparable to the standard antibiotic ampicillin.	[40]
*Acacia mearnsii* De Wild.	*Racosperma mearnsii* (De wild.) Pedley	Stem bark	Hexane, methanol, ethyl acetate, dichloromethane	*S*. *aureus*, *S*. *epidermidis*, *B. cereus*, *Micrococcus kristinae*, *S*. *faecalis*, *E*. *coli*, *Pseudomonas aeruginosa*, *Shigella flexneri*, *Klebsiella pneumonia*, *Serratia marcescens*	1–10 mg/mL	Hexane extract showed relatively weak activity against *S. aureus* and *E. coli* (MIC = 10 mg/mL), with no activity observed against *Shigella Flexneri* and *Klebsiella pneumonia*. Methanol extracts had moderate activity when tested against Gram-positive strains and *E. coli* (MIC = 5 mg/mL). The MIC values of ethyl acetate extracts ranged from 1 to 10 mg/mL against all tested bacterial strains.	[41]
*Senegalia burkei* Benth.	*Acacia burkei* (Benth.)	Bark	Dichloromethane: methanol, water	*Bacillus cereus*, *Enterococcus faecalis*, *Escherichia coli*, *Salmonella typhimurium*, *Shigella flexneri* and *Staphylococcus aureus*	0.25–3 mg/mL	Organic extract displayed noteworthy activity against *P. vulgarius* and *S. aureus*, 0.50 and 0.25 mg/mL, respectively. Water extracts showed mainly moderate activity (1–3 mg/mL) against tested strains with exception observed for *B. cereus* (0.75 mg/mL).	[42]
*Senegalia catechu* (L.f.)	*Acacia catechu* (L.f.)	Leaves	Methanol	*B*. *subtilis*, *S*. *aureus*, *S*. *typhi*, *E*. *coli*, *P*. *aeruginosa* and *Candida. albicans*	1–2 mg/mL	Leaf extract exhibited good activity against *S*. *typhi* (0.7 mg/mL), with moderate (1–2 mg/mL) microbial activity against the other tested microorganisms.	[43]
*Senegalia polyacantha* Willd.	*Acacia polyacantha* (Willd.)	Trunk bark	Ethanol, hydroethanol, aqueous	*S*. *aureus*, *E*. *coli*, *S*. *typhi*, *K*. *pneumoniae*	1.56–>100 mg/mL	Ethanolic stem bark extract displayed moderate activity against *S. aureus* (MIC = 1.56 mg/mL, MBC = 6.25 mg/mL) and *E. coli* (MIC = 3.13 mg/mL, MBC = 25 mg/mL), while the aqueous extract demonstrated weak activity against *S. typhi* with MIC and MBC values of 12.50 mg/mL and 100 mg/mL, respectively.	[44]
*Vachellia drepanolobium* Harms ex Y.sjostedt	*Acacia drepanolobium* Harms ex Y.sjostedt	Stem bark	Methanol	*E*. *coli*, *S*. *aureus*, *K*. *pneumoniae*, *P. aeruginosa*, *S*. *typhi*, *P*. *vulgaris*, *Candida albicans*	0.31–5 mg/mL	Stem bark extract exhibited good to moderate antimicrobial activity against all tested microorganisms with MIC ranging from 0.3125 to 5 mg/mL.	[45]
*Vachellia nilotica* (L.)	*Acacia nilotica* (L.)	Leaves Stem bark	Ethanol	*S*. *aureus*, *B*. *subtilis*, *P*. *aeruginosa*, *E*. *coli*	15.6–125 mg/mL	Ethanolic leaf extracts showed weak activity against the test pathogens, with MIC values ranging from 15.6 to 31.3 mg/mL and values at 31.3 mg/mL. For stem bark extracts, MIC was 125 mg/mL and MBC was 250 mg/mL for all test pathogens.	[46]
*Vachellia nilotica* (L.)	*Acacia nilotica* (L.)	Leaves	Ethanol	*Campylobacter coli*	3 mg/mL, 30 mg/mL and 70 mg/mL	Ethanolic leaf extract exhibited weak activity against *C. coli* with MIC value of 70 mg/mL.	[47]
*Vachellia nilotica* (L.)	*Acacia nilotica* (L.)	Bark, leaves	Acetone, water	*S*. *aureus*, *S*. *uberis*, *S. agalactiae*, *S*. *chromogenes*, *S*. *epidermidis*, *Klebsiella pnuemoniae*, *E*. *coli*, *Pseudomonas aeruginosa*, *P*. *vulgaris*, *E*. *aerogenes and Proteus mirabilis*	0.039 to 0.625 mg/mL	Acetone bark extract demonstrated noteworthy activity against all the tested bacterial strains (clinical isolates, ATCC and field strains) with an MIC ranging between 0.039 and 0.3130 mg/mL. Bark water extracts also showed strong activity amongst majority of the tested strains. Leaf extracts (acetone and water) showed good to moderate activity.	[48]
*Vachellia nilotica* (L.)	*Acacia nilotica* (L.)	Fruits	Methanol	*Salmonella typhi*, *P*. *aeruginosa*, *B. cereus*, *E*. *coli*, *K*. *pneumonia* and *Shigella flexneri*	100 mg/mL	Fruit extracts showed the largest inhibition zone against *S. typhi* (39 mm) and *B. cereus* (30 mm) at 100 mg/mL concentration relative to gentamicin, zone of inhibition (10 μg/disc).	[49]
*Vachellia nilotica* (L.)	*Acacia nilotica* (L.)	Leaves, pods, bark	Ethanol	*S*. *typhimurium*, *S*. *paratyphi*, *Salmonella* sp., *S*. *dysenteriae*, *Shigella* sp., and *S*. *flexneri*	50–200 mg/mL	Leaves, pods, bark displayed a synergistic effect when tested against multi-drug-resistant bacteria, with an inhibition zone ranging from 15 to 22.7 mm and MIC values between 100 and 200 mg/mL.	[50]
*Vachellia xanthophloea* Benth.	*Acacia xanthophloea* (Benth.) Banfi & Galasso	Leaf (Ag-NPs)	Aqueous	*Staphylococcus aureus*, *E*. *coli*		Silver nanoparticles showed excellent activity against *S. aereus* (MIC = 0.04 mg/mL, MBC = 0.04 mg/mL) and weak activity for *E. coli* (MIC = 0.33 mg/mL, MBC = 0.66 mg/mL).	[51]

Abbreviations: MIC—minimum inhibitory concentration, MBC—minimum bactericidal concentration.

**Table 3 plants-14-03107-t003:** In vitro and/or in vivo biological screening of *Acacia*/*Vachellia*/*Senegalia* species.

Bioassay	Plant Species (Accepted Name)	Plant Species (Synonym)	Plant Part	Extraction Solvent	Concentration Tested	Summary of the Tested Extracts	Reference
In vitro—Anthelmintic	*Senegalia senegal* (L.)	*Acacia senegal* (L.)	Stem bark	Methanol	0.25–1 mg/mL	Methanol stem bark extract demonstrated significant anthelmintic activity against *Fasciola gigantica*. The extract achieved 100% mortality at concentrations of 1 mg/mL and 0.5 mg/mL within 6 to 12 h, while 0.25 mg/mL resulted in 100% mortality within 24 h.	[52]
In vitro—Anthelmintic	*Vachellia xanthophloea* Benth.	*Acacia xanthophloea* (Benth.)	Bark	Methanol, cold water, boiled water	8, 16, 24, 32, 40% *v*/*v*	Methanolic extract demonstrated significant anthelmintic activity against L_3_ larvae of gastrointestinal nematodes in goats.	[53]
In vitro—Anti-inflammatory	*Vachellia Sieberiana* DC.	*Acacia Sieberiana* (DC.)	Leaves	Methanol, acetone	0.39–2 mg/mL	Methanol extracts inhibited NO (0.094 mg/mL) with the acetone extract exhibiting moderate inhibition of 15-LOX (IC_50_—2 mg/mL).	[54]
In vitro—Anti-inflammatory	*Vachellia farnesiana* (L.)	*Acacia farnesiana* (L.)	Leaves, bark	Ethanol	0.05 mg/mL	In most cases, bark extract inhibited COX-1 more than COX-2 enzyme, whereas leaf extract demonstrated higher anti-inflammatory activity against COX-2 relative to COX-1 enzyme.	[55]
In vitro—Antioxidant	*Senegalia ataxacantha* DC.	*Acacia ataxacantha* (DC.)	Bark	Hexane, methanol, dichloromethane, ethyl acetate, 70% ethanol	74.18 mg, GAE/100 mg, 23.14–26.65 mg QE/100 mg, 0.1 mg/mL	Ethyl acetate extract demonstrated antioxidant activity with DPPH radical scavenging inhibition of 92.62% and a FRAP value of 1273.63 µmol AAE/g.	[56]
In vitro—Antioxidant	*Vachellia karroo* Hayne.	*Acacia karroo* (Hayne.)	Leaves	Aqueous, acetone	0.2–1 mg/mL	Acetone and aqueous leaf extracts showed good antioxidant activity with observed IC_50_ values of 0.62 and 0.67 mg/mL (DPPH), 0.56 and 0.59 mg/mL (FRAP), as well as 0.43 and 0.60 mg/mL (NO), respectively.	[57]
In vitro—Antioxidant	*Vachellia eriolaba*	*Acacia eriolaba*	Leaves	Methanol, acetone	0.001–1 mg/mL	Acetone extract demonstrated strong antioxidant activity using the ABTS (0.001–0.14 mg/mL) and DPPH (0.003–1 mg/mL) assays.	[54]
In vitro—Antioxidant	*Acacia decurrens* Willd.	*Mimosa decurrens* J.C. Wendl.	Stem bark	Ethyl acetate, methanol	0.0125–0.05 mg/mL	The ethyl acetate and methanol fractions showed good antioxidant activity with IC_50_ values of 42.2–49.6 mg/mL for ABTS and 0.0378–0.075 mg/mL for DPPH compared to the standard ascorbic acid.	[40]
In vitro—Cytotoxicity	*Vachellia gerrardii* Benth.	*Acacia gerrardii* (Benth.)	Leaves	Methanol, acetone	0.039 mg/mL to 2.5 mg/mL	Methanol extract showed moderate cytotoxicity against RAW 264.7 macrophages with an LC_50_ value of 78.22 mg/mL.	[54]
In vitro—Cytotoxicity	*Vachellia nilotica* (L.)	*Acacia nilotica* (L.)	Bark, leaves	Acetone, water	0.0097 to >2.5 mg/mL	The cytotoxicity assays revealed that both bark and leaf extracts showed toxicity to mammalian cells, with bark extract showing a high level of toxicity (LC_50_ value of 0.029 mg/mL) and leaf extract displaying moderate toxicity with an LC_50_ value of 0.069 mg/mL.	[48]
In vitro—Cytotoxicity	*Vachellia eriolaba* E. Mey.	*Acacia eriolaba* E. Mey.	Leaves	Methanol, acetone	0.039 mg/mL to 2.5 mg/mL	Methanol extract demonstrated low cytotoxicity, >1000 µg/mL.	[54]
In vitro—Cytotoxicity	*Acacia mearnsii* De Wild.	*Racosperma mearnsii* (De Wild.) Pedley	Stem bark	Acetone	0.031–0.50 ug/mL	Extract showed non-toxic effects with LC_50_ > 0.1 mg/mL in the brine shrimp lethality assay.	[58]
In vitro—Cytotoxicity	*Vachellia nilotica* (L.)	*Acacia nilotica* (L.)	Pods with seeds	Ethanol	0.0101–0.016 mg/mL	Pod extract exhibited significant cytotoxicity with an LC_50_ value of 0.0101 mg/mL, indicating potential toxicity.	[59]
In vivo—Antioxidant	*Vachellia nilotica* (L.)	*Acacia nilotica* (L.)	Pods	Aqueous	1, 3, 5, 7.5, 10, 15 g/kg body weight	Aqueous extracts were reported to be toxic to broilers at the highest tested dose, 15 g/kg, causing liver damage.	[60]
In vivo—Anti-inflammatory	*Vachellia karroo* Hayne.	*Acacia karroo* (Hayne.)	Stem bark	Aqueous	100–200 mg/kg	Aqueous extract exhibited good anti-inflammatory and analgesic activities at doses of 100 and 200 mg/kg in the animal model.	[61]
In vivo—Toxicity	*Vachellia nilotica* (L.)	*Acacia nilotica* (L.)	Roots	Aqueous	50, 300, 500, 2000 mg/kg (acute), 125, 250, 500 mg/kg (sub-acute)	Aqueous extract was safe in single-dose administration but repeated doses above 250 mg/kg led to hepatotoxicity.	[62]
In vivo—Toxicity	*Vachellia nilotica* (L.)	*Acacia nilotica* (L.)	Leaves	Aqueous	250, 500, 1000 mg/kg	LD_50_ (3808 mg/kg) showed safety in acute exposure. The sub-acute exposure (28 days) at 500 and 1000 mg/kg caused mild hepatic and nephron toxicity.	[63]
In vivo—Toxicity	*Vachellia nilotica* (L.)	*Acacia nilotica* (L.)	Stem bark	Methanol	600, 800, 1000, 1200 mg/kg	Extract was safe at LD_50_ (1200 mg/kg) as no mortality was observed, and the animals did not exhibit signs of toxicity.	[64]
In vivo—Toxicity	*Vachellia sieberiana* DC.	*Acacia Sieberiana* (DC.)	Stem bark	Acetone	300, 600, 1200 mg/kg	No mortality or abnormal behaviour observed in rats at LD_50_ > 2000 mg/kg.	[65]
In vivo—Toxicity	*Senegallia polyacantha* (Willd.)	*Acacia polyacantha* Willd.	Leaves	Methanol	5000 mg/kg	LD_50_ > 5000 mg/kg showed no extract toxicity.	[66]
In vivo—Toxicity	*Senegalia ataxacantha* (DC.) Kyal. & Boatwr.	*Senegalia ataxacantha* (DC.) Kyal. & Boatwr.	Leaves	Methanol	50, 200, 400 mg/kg	Methanol extract was safe on acute exposure. However, prolonged use may produce harmful effects on the liver, kidney and stomach.	[67]

Abbreviations: DPPH—2,2-diphenyl-1-picrylhydrazyl, FRAP—ferric reducing antioxidant power, ABTS—2,2-azino-bis(3-ethylbenzothiazoline-6-sulfonic acid), TAC—total antioxidant capacity, NO—nitric oxide, 15-LOX—15-Lipoxygenases, COX—cyclooxygenase, HepG2—hepatocellular carcinoma, IC50—inhibitory concentration 50%, LD_50_—lethal dosage 50%, LDH—lactate dehydrogenase, ALP—alkaline phosphatase, ALT—alanie transaminase, AST—aspartae aminotransferase.

**Table 4 plants-14-03107-t004:** Phytochemical compounds isolated from *Acacia*/*Vachellia*/*Senegalia* species and their biological activity.

Plant Species	Synonym	Plant Part	Extraction Solvent	Isolated Compound	Method	Bioactivity	Reference
*Acacia saligna* (Labill.) H.L Wendl	*Mimosa Saligna* Labill.	Leaves	Methanol	(25S)-5b-spirostan-3β-yl-3-O-b-Dxylopyranosyl(1–3)-O-β-D-xylopyranosyl(1–4)-β-D-galactopyranoside, myricetin-3-O-rhamnoside (C7-O-C7) myricetin-3-O-rhamnoside, 3β-O-trans-pcoumaroyl-erythrodiol, quercetin-3-O-α-L-rhamnoside and myricetin-3-O-β-Lrhamnoside	TLC, UV, NMR	Antioxidant, Cytotoxicity	[68]
*Senegalia ataxacantha* (DC.) Kyal. & Boatwr.	*Acacia ataxacantha* DC.	Bark	Hexane, Dichloromethane, ethyl acetate, methanol	Lupeol, betulinic acid, betulinic acid-3-trans-caffeate	Column Chromatography (silica gel), TLC, HPLC, UV, NMR, Mass Spectrometry	Antimicrobial, Antioxidant	[17]
*Senegalia polyacantha* (Willd.) Seigler & Ebinger	*Acacia polyacantha* Willd.	Leaves	Methanol	Stigmasterol, β-Amyrin, 3-O-Methyl-D-chiro-inositol, Epicatechin, Quercetin-3-O-galactoside, 3-O-[β-D-xylopyranosyl-(1⟶4)-β-D-galactopyranosyl]-oleanolic acid, 3-O-[β-galactopyranosyl-(1⟶ 4)-β-D-galactopyranosyl]-oleanolic acid	Column chromatography (silica gel), TLC, Sephadex LH-20 chromatography, UV, NMR, Mass Spectrometry	Antibacterial	[69]
*Senegalia polyacantha* (Willd.) Seigler & Ebinger	*Acacia polyacantha* Willd.	Stem Bark	Methanol	Lupeol, 2,3-Dihydroxypropyltetracosanoate, Methyl Gallate	Column chromatography (silica gel), TLC, Sephadex LH-20 chromatography, UV, NMR, Mass Spectrometry	Antibacterial	[69]
*Senegalia polyacantha* (Willd.) Seigler & Ebinger	*Acacia polyacantha* Willd.	Leaves	Methanol	Stigmasterol, β-Amyrin, 3-O-β-D-glucopyranosylstigmasterol, 3-O-methyl-D-chiro-inositol, Epicatechin, Quercetin-3-O-glucoside, 3-O-[β-D-xylopyranosyl-(1⟶4)-β-D-galactopyranosyl]-oleanolic acid, 3-O-[β-galactopyranosyl-(1⟶4)-β-D-galactopyranosyl]-oleanolic acid	Column chromatography (silica gel), TLC, Sephadex LH-20 chromatography, UV, NMR, Mass Spectrometry	Antibacterial	[70]
*Vachellia xanthophloea* (Benth.) Banfi & Galasso	*Acacia xanthophloea* Benth.	Leaves	Dichloromethane/methanol (1:1)	Methyl Gallate, 3-O-Methylquercetin, Kaempferol, Apigenin, Pinoresinol, Lupeol, Phytol	Column chromatography (silica gel), TLC, Sephadex LH-20 chromatography, HPLC, UV, NMR	Antiplasmodial	[71]

HPLC—high-performance liquid chromatography, NMR—nuclear magnetic resonance, UV—ultraviolet spectroscopy, TLC—thin layer chromatography, MS—mass spectrometry.

## Data Availability

No new data were created or analyzed in this study.

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
