# Peer review of "Ethnoveterinary Potential of Acacia (Vachellia and Senegalia) Species for Managing Livestock Health in Africa: From Traditional Uses to Therapeutic Applications"

_plants, 2025, doi:10.3390/plants14193107_

Round 1

Reviewer 1 Report

Comments and Suggestions for Authors

Dear authors,

I have reviewed your review “Ethnoveterinary potential of Acacia (Vachellia and Senegalia) species for managing livestock health in Africa: From traditional uses to therapeutic applications”.

The review is very well written and illustrated. Review falls within the scope of the journal and the subject matter is quite interesting. Therefore, I recommend accepting this review in its present form.

Author Response

Comment: I have reviewed your review “Ethnoveterinary potential of Acacia (Vachellia and Senegalia) species for managing livestock health in Africa: From traditional uses to therapeutic applications”.

The review is very well written and illustrated. Review falls within the scope of the journal and the subject matter is quite interesting. Therefore, I recommend accepting this review in its present form.

Response: We are grateful to the reviewer for the positive comments about the review.

Reviewer 2 Report

Comments and Suggestions for Authors

The manuscript “Ethnoveterinary potential of Acacia (Vachellia and Senegalia) species for managing livestock health in Africa: From traditional uses to therapeutic applications” presents a systematic review of 43 eligible studies published between 2001 and 2024. The authors synthesize ethnoveterinary knowledge, pharmacological evidence, and safety data related to Acacia/Vachellia/Senegalia species across Africa. Key contributions include mapping ethnoveterinary practices by country, identifying the most frequently used species and plant parts, and summarizing antimicrobial, anti-inflammatory, antioxidant, and anthelmintic activities. The paper’s main strength lies in its comprehensive scope, bridging traditional knowledge and modern pharmacological evidence, and highlighting the potential integration of Acacia species into livestock health management while also pointing out toxicity concerns and research gaps.

1. Could you provide more detail on the exact inclusion and exclusion criteria used during the literature screening (beyond "non-animal studies and reviews")? This would strengthen reproducibility. Providing the exact search string would give transparency.

2. Since Acacia species have undergone major reclassification, how did you handle synonyms across older studies to ensure consistency? Did you exclude any studies due to unclear taxonomy?

3. The discussion could benefit from a comparison between dosages traditionally reported in ethnoveterinary practice and those tested in toxicological/pharmacological assays. Could you integrate this to strengthen the practical relevance?

4. As the review concludes, in vivo and clinical trials are limited. Could you highlight specific priority areas (e.g., diseases, species, or compound groups) where such studies are most urgently needed?

5. Since Acacia species are ecologically important, could you briefly discuss the sustainability of harvesting practices and the potential need for conservation strategies when recommending wider veterinary use?

Author Response

Comment 1:

  • Could you provide more detail on the exact inclusion and exclusion criteria used during the literature screening (beyond "non-animal studies and reviews")? This would strengthen reproducibility. Providing the exact search string would give transparency.

Response:

  • Thank you for this important suggestion. We have now included a more detailed account of our inclusion and exclusion criteria in the Methods section (lines 583 – 585). We have also included the exact Boolean search strings used to enhance reproducibility (lines 568 – 571).

Comment 2:

  • Since Acacia species have undergone major reclassification, how did you handle synonyms across older studies to ensure consistency? Did you exclude any studies due to unclear taxonomy?

Response:

  • We appreciate this insightful point. As highlighted in lines 585–591, we cross-referenced all species names cited in older studies with the updated taxonomic classifications available through Plants of the World Online (POWO). For consistency, both the original names and their currently accepted synonyms have been included throughout the review. No studies were excluded based on outdated nomenclature, provided the identity of the plant species could be reliably confirmed.

Comment 3:

  • The discussion could benefit from a comparison between dosages traditionally reported in ethnoveterinary practice and those tested in toxicological/pharmacological assays. Could you integrate this to strengthen the practical relevance?

Response:

  • This is a valuable recommendation. Unfortunately, a substantial gap exists in the literature regarding the quantification of dosages used in traditional ethnoveterinary practices. Many ethnobotanical reports mention the use of Acacia species but fail to provide measurable doses or concentrations. This gap is now discussed in the revised manuscript, emphasizing the need for future studies to document dosage-related data (lines 487 – 498).

Comment 4:

  • As the review concludes, in vivo and clinical trials are limited. Could you highlight specific priority areas (e.g., diseases, species, or compound groups) where such studies are most urgently needed?

Response:

  • Thank you for the suggestion. We now highlight priority research areas in lines 617 –621.

Comment 5:

  • Since Acacia species are ecologically important, could you briefly discuss the sustainability of harvesting practices and the potential need for conservation strategies when recommending wider veterinary use?

Response

  • We fully agree. A brief discussion on the ecological sustainability of Acacia harvesting practices and potential conservation strategies has now been incorporated in lines 621–627.

Reviewer 3 Report

Comments and Suggestions for Authors

In the manuscript "Ethnoveterinary potential of Acacia (Vachellia and Senegalia) species for managing livestock health in Africa: From traditional uses to therapeutic applications”, Msimango et al summarized the traditional and therapeutic uses of Acacia species (Vachellia and Senegalia) in ethnoveterinary medicine across Africa, including Ethiopia, Nigeria, South Africa, and Namibia, with high usage. This manuscript involves in application of these plants in livestock health and worth of publication. However, there are many reviews with the ethnopharmaceutical potential in Pubmed. The phytochemistrial detail and analyses against pathogens and diseases as well as the possible targets should be summarized and compared clearly for the reader. In addition, the toxicity should be also discussed with figures and tables. The references should be updated and followed mdpi format. Overall, this reviewer thinks that this manuscript can be acceptable for publication in Plants.

Author Response

Comment 1:

  • The phytochemistrial detail and analyses against pathogens and diseases as well as the possible targets should be summarized and compared clearly for the reader.

Response

  • We have revised the manuscript to provide a clearer summary of phytochemical classes and their associated bioactivities in lines 539 – 553.

Comment 2:

  • The toxicity should be also discussed with figures and tables.

Response:

  • Thank you for this valuable suggestion. However, we would like to clarify that the toxicity studies including the test models, doses administered, and observed effects are already presented in Table 3, titled "In vitro and/or in vivo biological screening of Acacia/Vachellia/Senegalia". This table summarizes both pharmacological and toxicological findings to provide a comprehensive overview of the biological activity of the included species.

Comment 3:

  • The references should be updated and followed mdpi format.

Response:

We are grateful for your positive evaluation. All references have been reviewed and updated to conform to MDPI’s citation style.